# CENP-C functions in centromere assembly, the maintenance of CENP-A asymmetry and epigenetic age in Drosophila germline stem cells

**Ben L. Carty**, **Anna A. Dattoli**¤, **Elaine M. Dunleavy**\*

Centre for Chromosome Biology, Biomedical Sciences, National University of Ireland Galway, Galway, Ireland

¤ Current address: Perelman School of Medicine, University of Pennsylvania, Philadelphia, United States of America

\* elaine.dunleavy@nuigalway.ie

**Data Availability Statement:** All relevant data are within the paper and numerical data that underlies graphs or summary statistics is provided in the Supporting Information file (S1 Data.xml).

## Abstract

Germline stem cells divide asymmetrically to produce one new daughter stem cell and one daughter cell that will subsequently undergo meiosis and differentiate to generate the mature gamete. The silent sister hypothesis proposes that in asymmetric divisions, the selective inheritance of sister chromatids carrying specific epigenetic marks between stem and daughter cells impacts cell fate. To facilitate this selective inheritance, the hypothesis specifically proposes that the centromeric region of each sister chromatid is distinct. In Drosophila germ line stem cells (GSCs), it has recently been shown that the centromeric histone CENP-A (called CID in flies)—the epigenetic determinant of centromere identity—is asymmetrically distributed between sister chromatids. In these cells, CID deposition occurs in $G_2$ phase such that sister chromatids destined to end up in the stem cell harbour more CENP-A, assemble more kinetochore proteins and capture more spindle microtubules. These results suggest a potential mechanism of 'mitotic drive' that might bias chromosome segregation. Here we report that the inner kinetochore protein CENP-C, is required for the assembly of CID in $G_2$ phase in GSCs. Moreover, CENP-C is required to maintain a normal asymmetric distribution of CID between stem and daughter cells. In addition, we find that CID is lost from centromeres in aged GSCs and that a reduction in CENP-C accelerates this loss. Finally, we show that CENP-C depletion in GSCs disrupts the balance of stem and daughter cells in the ovary, shifting GSCs toward a self-renewal tendency. Ultimately, we provide evidence that centromere assembly and maintenance via CENP-C is required to sustain asymmetric divisions in female Drosophila GSCs.

## Author summary

Stem cells can divide in an asymmetric fashion giving rise to two daughter cells with different fates. One daughter remains a stem cell, while the other can differentiate and adopt

**Funding:** E.M.D. is funded by Science Foundation Ireland -PIYRA 13/YI/2187 (www.sfi.ie). A.A.D. was funded by a Government of Ireland Postdoctoral Fellowship 2017/1324 from the Irish Research Council (www.research.ie) and Science Foundation Ireland-PIYRA 13/YI/2187 awarded to E.M.D. B.L.C. is funded by a Government of Ireland Postgraduate Fellowship 2018/1208 from the Irish Research Council and by Science Foundation Ireland-PIYRA 13/YI/2187 awarded to E.M.D. The funders had no role in study design, data collection and analysis, decision to publish, or preparation of the manuscript.

**Competing interests:** The authors have declared that no competing interests exist.

a new cell fate. Germline stem cells in the testes and ovaries give rise to differentiating daughter cells that eventually form the gametes, eggs and sperm. Here we investigate mechanisms controlling germline stem cell divisions occurring in the ovary of the fruit fly *Drosophila melanogaster*. Centromeres are epigenetically specified loci on chromosomes that make essential connections to the cell division machinery. Our study is focused on the centromere component CENP-C. We show that CENP-C is critical for the correct assembly of centromeres that occurs prior to cell division in germline stem cells. In addition, we find that CENP-C is asymmetrically distributed between stem and daughter cells, with more CENP-C at stem cell centromeres. Finally, we show that CENP-C depletion in germline stem cells disrupts the balance of stem and daughter cells in the developing ovary, impacting on cell fate. Taken together, we propose that CENP-C level and function at centromeres plays an important role in determining cell fate upon asymmetric division occurring in stem cells.

## Introduction

Stem cells are unique in that these cells can divide to give rise to daughter cells of different fates. Stem cells can undergo two distinct mitotic division types; symmetric cell division (SCD) in which the stem cell self-renews, and asymmetric cell division (ACD) in which the stem cell produces one daughter cell that undergoes differentiation [1,2]. Misregulation of the balance between SCD and ACD can lead to diseases, such as cancer and infertility [3–5]. Stem cell divisions are regulated by cell extrinsic means, with well-characterised roles for signaling pathways such as Wnt, fibroblast growth factor (FGF), bone morphogenetic (BMP) [6,7]. Additionally, epigenetic mechanisms at the level of chromatin, histones and associated modifications have been implicated in the regulation of ACD. Studies in Drosophila germ line stem cells showed that prior to ACD, parental histones H3 and H4 [8,9], as well as histone H3 phosphorylated at position threonine 3 [10], are enriched on chromosomes that end up in the future stem cell. The differential distribution of histones H3 and H4 has recently been reported also in mouse embryonic stem cells [11]. These observations are in line with the 'silent sister' hypothesis, which proposed that sister chromatids–each carrying distinct epigenetic marks that result in differential gene expression—are selectively inherited between stem and daughter cells [12]. Moreover, the hypothesis suggested that the centromeres of each sister chromatid would also be distinct in order to facilitate selective chromosome segregation [12].

Centromeres are the chromosomal loci that specify the site of kinetochore assembly and microtubule attachment, playing a critical role in orchestrating chromosome segregation in cell division [13,14]. This locus is epigenetically defined by the incorporation of the histone H3 variant CENP-A, which is both necessary and sufficient for centromere specification and function [15–17]. Each cell cycle, newly synthesized CENP-A is assembled at centromeres to ensure functional centromere maintenance [18,19]. Recently, it has been shown in both Drosophila male and female germ line stem cells (GSCs) that CID assembly shows unique properties [20,21]. Firstly, CID is deposited at centromeres prior chromosome segregation, during $G_2$/prophase, a cell cycle time that is distinct compared to symmetrically diving cells [20,21]. Secondly, CID is unevenly distributed between sister centromeres, with between 1.2–1.5 fold more CID inherited by 'stem' side sister chromatids [20,21]. A third line of evidence showed that parental CID–as opposed to newly synthesized CID—is found to be enriched in both intestinal and germline stem cells [21,22]. Finally, studies in GSCs showed that the mitotic spindle is asymmetric both temporally and with respect to the distribution of microtubules; at

prometaphase sister chromatids of the future stem cell attach first to the spindle and more spindle microtubule are observed in the stem cell side at metaphase [20,21]. Taken together, these studies propose a model by which CID asymmetry can drive the selective attachment of microtubules leading to the non-random segregation of sister chromatids [23,24].

Further investigations into how the mitotic chromosome segregation machinery—and specifically centromeres—are altered in asymmetric divisions are now needed. Indeed, relatively little is known about centromere assembly and maintenance in stem cells. In addition to CID, the Drosophila centromeric core is comprised of two key components, the inner kinetochore protein CENP-C and the centromere assembly factor CAL1 [25,26]. CAL1 binds to CID-H4 dimers and assembles CID nucleosomes [27–29]. CENP-C binds to CID containing nucleosomes, and also interacts directly with CAL1, recruiting new CAL1-CID-H4 to the centromere [28–30]. In addition, CAL1 can then recruit new CENP-C to the centromere, closing the epigenetic loop [28,29]. In Drosophila GSCs, both CAL1 and CENP-C are asymmetrically distributed between stem and daughter cells [20,21]. Functional experiments–either overexpression or depletion—have shown that CAL1 is required to maintain CID asymmetry in GSCs, impacting on cell fate and development [20,21]. CENP-C is also critical for the assembly and maintenance of CID/CENP-A at fly and human centromeres [25,31,32]. Yet, whether CENP-C can regulate stem cell asymmetric division beyond its canonical mitotic kinetochore function remains unclear. In this study, we investigate CENP-C function in Drosophila GSCs. We find that CENP-C is required for CID assembly in GSCs, as well as maintaining appropriate CID asymmetry between stem and daughter cells. In addition, we determine CID and CENP-C levels to decrease in accordance with GSC age. We propose that CENP-C's function in CID assembly and asymmetry maintains the balance of symmetric and asymmetric divisions in the GSC niche impacting on long term GSC maintenance in the ovary.

## Results

### CENP-C is assembled at GSC centromeres in $G_2$/prophase

At the apical end of the Drosophila germarium (Fig 1A), 2–3 GSCs are found attached to cap cells (Fig 1B). Female GSCs divide asymmetrically to give a differentiating daughter cell called a cystoblast (CB) and another GSC [33]. We previously showed that CID is assembled at GSC centromeres between the end of DNA replication up until at least prophase [20]. To assess the cell cycle timing of CENP-C assembly in GSCs, we used 5-ethynyl-2′-deoxyuridine (EdU) incorporation to mark cells in and out of S-phase and 1B1 staining to mark the spectrosome, the shape of which can be used to define the cell cycle stage [34,35] (Fig 1C–1G'). As previously described [20], GSCs in mid to late S-phase show a pan nuclear EdU staining pattern, in which the spectrosome forms a bridge shape (Fig 1C–1G). GSCs that were EdU negative with a round spectrosome and with centromeres distributed throughout the nucleus, but without condensed chromosomes, were deemed to be in $G_2$/prophase (Fig 1C'–1G'). We then quantified total CENP-C fluorescent intensity (integrated density) at centromeres at both stages (Fig 1H). We found an increase in CENP-C level in $G_2$/prophase cells compared to S-phase cells. Quantitation revealed an average increase of 38% in CENP-C (S-phase = 23.36±1.84, n = 32 cells; $G_2$/prophase = 32.14±1.611, n = 34 cells). These results indicate that similar to CID, CENP-C is assembled at GSCs centromeres between the end of S-phase and $G_2$/prophase.

### CENP-C is required for CID assembly in the germline, specifically in GSCs

To determine if CENP-C is required for CID assembly in GSCs, we used the GAL4-UAS system to induce the RNAi-mediated depletion of CENP-C using the germline-specific driver *nanos-GAL4*. To confirm CENP-C knock down, control *nanos-GAL4* and CENP-C RNAi

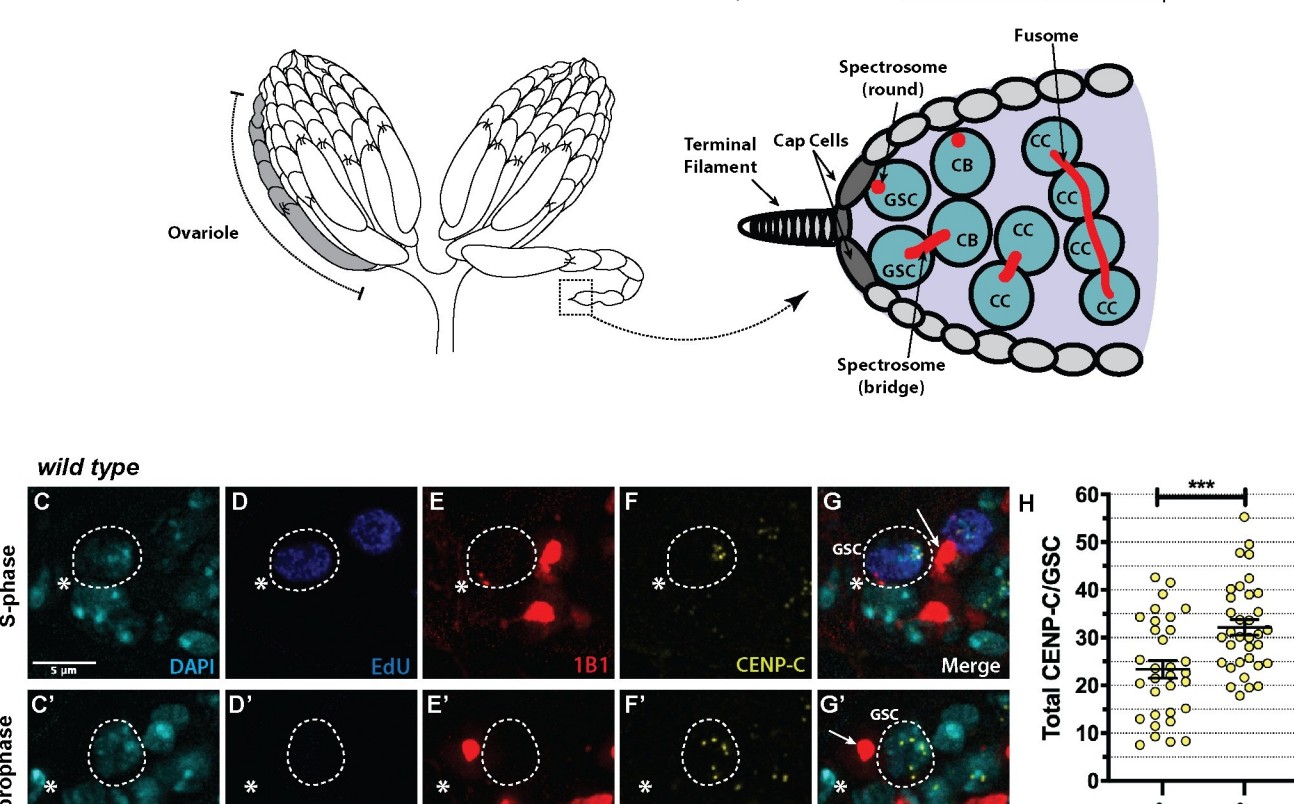

**Fig 1. CENP-C is assembled between S-phase and G$_2$/prophase in female GSCs.** (A) Schematic of the Drosophila ovary (created by B. L. Carty), composed of 16 ovarioles (one ovariole is highlighted in grey) organised into developing egg chambers. The GSC niche is located in the anterior-most chamber of the ovariole, the germarium (boxed). (B) Schematic of the GSC niche and 2- and 4-cell cysts in the germarium. G$_2$/prophase GSCs can be identified with a round spectrosome attached to the cap cells. CB = cystoblast, CC = cystocyte. (C-G') Immunofluorescent image of a wild type GSC (circled) in S-phase with a bridged spectrosome (white arrow) (G) and in G$_2$/prophase with a round spectrosome (white arrow) (G') stained with DAPI (cyan), EdU (blue), spectrosome (1B1, red) and CENP-C (yellow). The circled GSC is a projection of z-stacks that displays the spectrosome morphology (round or bridged) and all centromere foci of that specific cell. (H) Quantitation of total CENP-C fluorescent intensity (integrated density) in GSCs at S-phase and G$_2$/prophase. ***p<0.001. Scale bar = 5 μm. Error bars = Standard Error of the Mean (SEM).

ovaries were stained with antibodies against CENP-C and 1B1 to mark the spectrosome (S1A–S1D' Fig). We quantified the total CENP-C fluorescent intensity (integrated density) in GSCs with a round spectrosome, indicative of cells in G$_2$/prophase (S1E Fig). Quantitation revealed an approximate 60% depletion of CENP-C in GSCs (*nanos-GAL4* = 34.45±1.65, n = 29 cells; CENP-C RNAi = 13.70±1.63, n = 28 germaria). We next labelled and quantified CID fluorescent intensity in GSCs depleted for CENP-C, both at S-phase and at G$_2$/prophase (Fig 2A–2J'). As expected in the *nanos-GAL4* control, CID intensity increased between S-phase and G$_2$/prophase (S-phase = 15.82±0.73, n = 40 cells; G$_2$/prophase = 24.58±1.45, n = 43 cells), by approximately 35% on average (Fig 2K). However, in the CENP-C RNAi we did not observe this increase (S-phase = 17.46±1.06, n = 36 cells; G$_2$/prophase = 15.50±0.96, n = 43 cells) (Fig 2K). Indeed, CID levels were comparable between S-phase and G$_2$/prophase. This result indicates that CENP-C is specifically required for CID assembly that occurs between S-phase and prophase in GSCs.

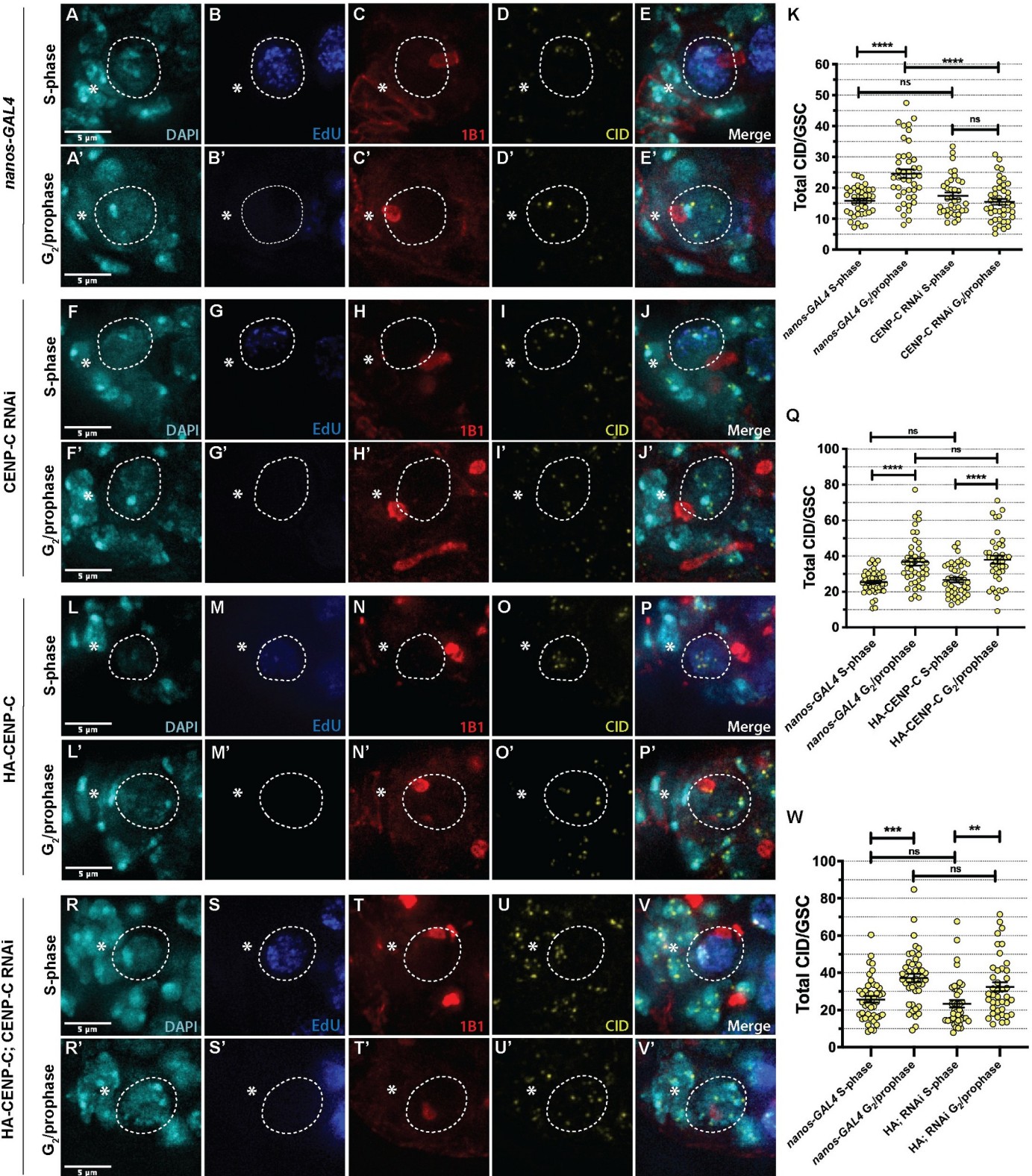

**Fig 2. CENP-C is required for CID assembly in GSCs.** (A-E') nanos-GAL4, (F-J') CENP-C RNAi, (L-P') HA-CENP-C and (R-V') HA-CENP-C; CENP-C RNAi (rescue) stained with DAPI (cyan), EdU (blue), 1B1 (red) and CID (yellow). S-phase GSCs (A-E, F-J, L-P, R-V) are positive for EdU, contain a bridge spectrosome and clustered centromeres. $G_2$/prophase GSCs (A'-E', F'-J', L'-P', R'-V') are EdU negative, contain a round spectrosome and dispersed centromeres. The circled

GSC is a projection of z-stacks that displays the spectrosome morphology (round or bridged) and all centromere foci of that specific cell. * denotes cap cells. Scale bar = 5 $\mu$m. Quantitation of total CID fluorescent intensity (integrated density) in GSCs at S-phase and $G_2$/prophase in nanos-GAL4 and (K) CENP-C RNAi, (Q) HA-CENP-C and (W) HA-CENP-C; CENP-C RNAi. ***p<0.001, **p<0.01, ns = non-significant. Error bars = SEM.

We next investigated whether the localisation of the CID assembly factor CAL1 was affected by CENP-C knockdown. For this, we antibody-stained control and CENP-C-depleted germaria for CAL1, as well as CENP-C in order to distinguish centromeric from nucleolar CAL1 (S2A–S2C' Fig). Both centromeric and nucleolar CAL1 was visible in the *nanos-GAL4* and CENP-C RNAi. Using residual CENP-C signals to mark centromeres, we quantified total centromeric CAL1 in GSCs at $G_2$/prophase and found it to be reduced in the CENP-C RNAi compared to the *nanos-GAL4* control (*nanos-GAL4* = 16.01±1.45, n = 30 cells; CENP-C RNAi = 9.44±0.85, n = 29 germaria) (S2D Fig). This result is in line with the structural evidence that CENP-C is the recruitment factor for CAL1-CID-H4 complexes, marking the centromere for new CID assembly [28]. Lastly, to assess if CENP-C is required for CID localisation at later stages of development, we knocked down CENP-C using the *bam-GAL4* driver active in 4–8 cell cysts. As previously reported [20], staining for CENP-C revealed an effective knock down at this stage and development of germaria appeared normal (S2E–S2H' Fig). Surprisingly, we noted that at this stage knockdown of CENP-C did not lead to a major reduction in CID level (S2I–S2L' and S2N–S2Q' Fig). Using phosphorylation at serine 10 of histone H3 (H3S10P) staining to identify synchronously dividing 8-cell cysts in mitosis, we quantified either total CENP-C or CID level per nucleus. In the CENP-C RNAi, we confirmed a 40% reduction in CENP-C (S2M Fig), however no significant change in CID intensity was measured between the knockdown and *bam-GAL4* control (S2R Fig). This finding is comparable to our previous observations for CID and CAL1 [20], and indicates that a 40% reduction in CENP-C does not alter CID assembly in later divisions occurring in the germarium.

## Excess CENP-C does not promote additional CID assembly in GSCs

To further monitor CENP-C function in CID assembly in GSCs, we overexpressed an N-terminal HA-tagged CENP-C using the *nanos-GAL4* driver (S1F–S1I Fig). To measure the extent of CENP-C over-expression, we labelled total CENP-C (tagged and endogenous) with an anti-CENP-C antibody and quantified its intensity in *nanos-GAL4* and HA-CENP-C GSCs at $G_2$/prophase (S1J Fig). Here, total centromeric CENP-C level increased by approximately 45% (*nanos-GAL4* = 31.86±2.59, n = 22 cells; HA-CENP-C = 51.02±4.49 n = 20 cells). We then measured CID assembly between S-phase and $G_2$/prophase in the background of increased CENP-C (Fig 2L–2P'). In GSCs overexpressing HA-CENP-C, CID intensity increased at the expected rate between S-phase and $G_2$/prophase, in line with the *nanos-GAL4* driver (*nanos-GAL4$_{S-phase}$* = 25.44±0.88, n = 48 cells; *nanos-GAL4$_{G2/prophase}$* = 36.77±2.01, n = 45 cells; HA-CENP-C$_{S-phase}$ = 26.63±1.28, n = 46 cells; HA-CENP-C$_{G2/prophase}$ = 37.99±2.20, n = 41 cells) (Fig 2Q). Moreover, fluorescence values between control and HA-CENP-C are comparable, indicating that increased CENP-C level does not correlate with increased CID assembly in GSCs. We next designed rescue experiments, in which we overexpressed HA-CENP-C that is resistant to the shRNA in the CENP-C RNAi background (Figs 2R–2V' and S1A"–S1D"). Upon over-expression, we quantified total CENP-C levels, comparing the HA-CENP-C; CENP-C RNAi to that of *nanos-GAL4* and CENP-C RNAi (S1E Fig). Here, 'rescued' GSCs displayed an 85% restoration of total CENP-C levels (*nanos-GAL4* = 34.45±1.65, n = 29 cells; CENP-C RNAi = 13.70±1.63, n = 28 germaria; HA-CENP-C; CENP-C RNAi = 29.49±2.09, n = 28 germaria). Finally, we measured CID assembly between S-phase and $G_2$/prophase in the HA-CENP-C; CENP-C RNAi background (Fig 2W). In this case, CID assembly was

partially rescued, displaying an increase in CID level from S-phase to $G_2$/prophase, although not quite to the CID level in the control (*nanos-GAL4*$_{S\text{-}phase}$ = 25.68±1.76, n = 43 cells; *nanos-GAL4*$_{G2/prophase}$ = 37.24±1.98, n = 44 cells; HA-CENP-C;CENP-C RNAi$_{S\text{-}phase}$ = 23.37±1.98, n = 42 cells; HA-CENP-C;CENP-C RNAi$_{G2/prophase}$ = 32.51±2.47, n = 41 cells). These results show that over-expression of CENP-C alone does not affect CID assembly, but CENP-C expression rescues the defect in CID assembly observed in the CENP-C RNAi.

## Reduced CENP-C increases CID asymmetry between GSCs and CBs

Our previous characterisation of centromere positioning in GSCs and CBs at anaphase and DNA replication, allowed us to conclude that both cells enter synchronously into S-phase immediately at the end of mitosis, without a detectable $G_1$ phase [20]. We also showed that in addition to CID, CENP-C is asymmetrically distributed between GSC-CB S-phase 'pairs' [20]. Using the H3S10P marker we could also confirm that CENP-C is asymmetrically distributed (approximately 1.4 fold) between GSCs and CBs in mitosis, at very early anaphase and at telophase (Fig 3A–3D'). In S-phase, we again confirmed 1.2 fold asymmetry for CID in *nanos-GAL4* (Fig 3E and 3I) and then tested if CENP-C is required for the asymmetric distribution of CID. For this, we measured CID intensity in GSC-CB S-phase pairs, expressed as a ratio of total CID in GSC/CB, in CENP-C-depleted GSCs compared to the control *nanos-GAL4* (Figs 3F and S3A). Quantitation revealed a significant increase in the GSC/CB ratio of CID intensity to 1.44 in the CENP-C RNAi versus 1.2 in controls (*nanos-GAL4* $_{GSC/CB}$ = 1.19±0.06, n = 40 cells; CENP-C RNAi$_{GSC/CB}$ = 1.44±0.08, n = 36 cells (Fig 3I). This indicates that in addition to CID assembly in $G_2$/prophase, CENP-C potentially functions in maintaining CID asymmetry in S-phase.

We next investigated CID asymmetry upon HA-CENP-C overexpression (Fig 3G). Comparing the ratio of total CID in GSC-CB pairs in S-phase, quantitation showed no significant change asymmetry (Figs 3J and S3B). In this case, HA-CENP-C overexpression did not significantly affect the GSC/CB ratio (*nanos-GAL4*$_{GSC/CB}$ = 1.22±0.06, n = 47 cells; HA-CENP-C$_{GSC/CB}$ = 1.12±0.04, n = 46 cells) (Fig 3J). To verify that the shift in CID asymmetry to 1.44 was dependent on CENP-C, we performed the same analysis in the HA-CENP-C; CENP-C RNAi rescue line (Fig 3H). Indeed, quantitation of rescue versus *nanos-GAL4* controls returned the expected CID ratio of 1.2 (*nanos-GAL4*$_{GSC/CB}$ = 1.23±0.07, n = 43 cells; HA-CENP-C; CENP-C RNAi$_{GSC/CB}$ = 1.20±0.06, n = 42 cells) (Figs 3K and S3C). Taken together, these results show that at 5 days old although CID asymmetry is perturbed after CENP-C RNAi, supply of excess CENP-C is not sufficient to drive CID asymmetry in stem and daughter cells at S-phase.

## CENP-C regulates GSC proliferation and long term GSC maintenance

To probe the function of CENP-C in GSC proliferation or maintenance, control and CENP-C knockdown ovaries were stained for the germ cell marker VASA, as well as 1B1 marking round spectrosomes and branched fusomes. Control *nanos-GAL4* contained the expected GSC and germ cell content, in line with previous studies [20]. Different from previous CID and CAL1 knockdowns that resulted in empty germaria [20], CENP-C depleted germaria revealed a spectrum of germ cell proliferation phenotypes (Fig 4A–4D'). Previously, GSC loss and complex differentiation phenotypes were described for CENP-C depletion in a large-scale RNAi screen performed in female GCSs [36]. Despite defective germaria, we observed that egg chamber development and the production of mature eggs continued in the CENP-C RNAi (Fig 4E–4H). Quantitation of phenotypes (Fig 4I) showed that over one third of germaria (35%) analysed 5 days after eclosion showed normal development, comparable to the control. However,

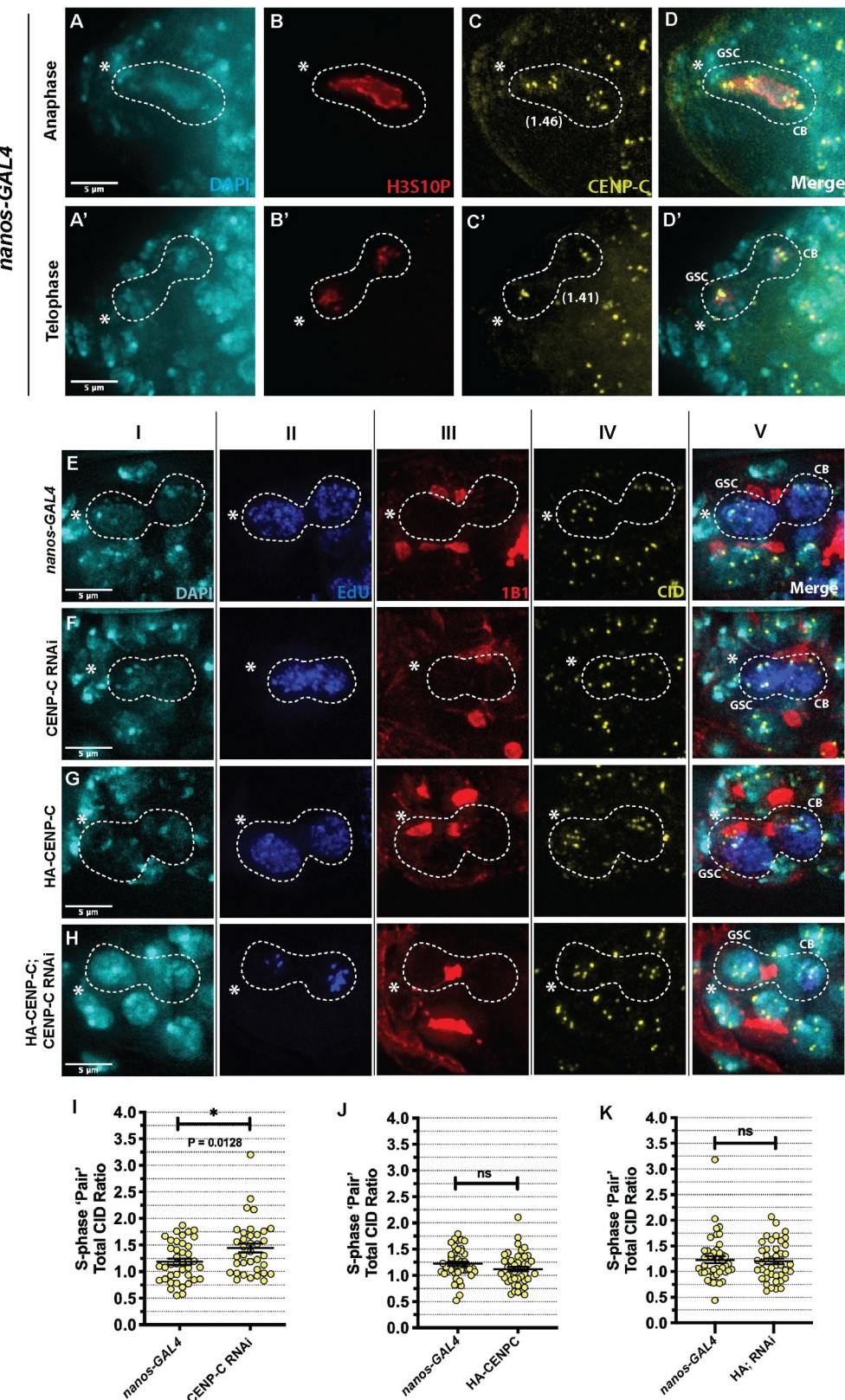

**Fig 3. CENP-C is asymmetrically distributed in mitosis and its depletion enhances the asymmetric CID distribution between GSCs and CBs at S-phase.** Control (nanos-GAL4) GSC at anaphase (A-D) and telophase (A'-

D') of mitosis stained for H3S10P (red), CENP-C (yellow), and DAPI (cyan). Values indicate fold differences in CENP-C intensities between GSC and CB centromeres. (EI-V) nanos-GAL4, (FI-V) CENP-C RNAi, (GI-V) HA-CENP-C and (HI-V) HA-CENP-C; CENP-C RNAi (rescue) stained with DAPI (cyan), EdU (blue), 1B1 (red) and CID (yellow). S-phase GSCs and CBs are positive for EdU, contain a bridge spectrosome and clustered centromeres. Dashed white line outlines GSC/CB pairs. Images are projections of z-stacks that display the bridged spectrosome morphology and all centromere foci of each GSC/CB pair. * denotes cap cells. Scale bar = 5 $\mu$m. Quantitation of the ratio of total CID fluorescent intensity (integrated density) between GSC/CB S-phase pairs in nanos-GAL4 and (I) CENP-C RNAi, (J) HA-CENP-C and (K) HA-CENP-C; CENP-C RNAi (rescue). Each point represents the ratio of total CID between GSC versus its corresponding CB. ns = non-significant. *p<0.05. Error bars = SEM.

another third (32%) displayed an accumulation of germ cells, indicative of a proliferation defect consistent with germ line tumours [37]. The final third (29%) displayed isolated GSC and CBs located in the niche and 4–8 cell cyst stages were lacking, indicative of a differentiation defect. Finally, a small proportion of germaria (4%) lacked GSCs entirely. Analysis of germaria 10 days after eclosion revealed an exacerbation of the GSC loss phenotype (21%) possibly due to further CENP-C depletion (Fig 4I). Importantly, HA-CENP-C overexpression almost completely rescued the differentiation defect and GSC loss phenotypes at 5 days, when expressed in conjunction with the CENP-C shRNA (Fig 4I). These results suggest that CENP-C is required for GSC proliferation, as well as the long-term maintenance of the GSC population. Notably, HA-CENP-C over-expression did not rescue the germ line tumour phenotype, possibly indicating that excess CENP-C or the incorrect timing of CENP-C expression or turnover can lead to proliferation defects. Indeed, knockdown of CENP-C at the adult stage using the temperature sensitive *tubulin-GAL80* driver in combination with *nanos-GAL4* resulted in germaria mostly displaying germline tumour defects (S3D-K' Fig). We also cannot exclude the possibility that the rescue was incomplete as the HA-CENP-C protein is not fully functional.

We next investigated if the accumulation of germ cells after CENP-C depletion might be due to a cell cycle block or delay. Given that CENP-C normally functions in kinetochore attachment to microtubules, we assayed whether cells were blocked in mitosis using H3S10P to mark cells at late $G_2$-phase and mitosis. CENP-C depleted germaria displaying the germline tumour phenotype are generally negative for H3S10P (S4A–S4D' Fig) and we did not observe a change in the GSC mitotic index (S4E Fig) nor in the number of mitotic cysts per germaria (S4F Fig). Moreover, the kinetochore protein Spc105 localised as expected at prometaphase [38,39], although at a reduced level (S4G–S4K' Fig). Finally, no significant change in centromere number (either of CID or Cen3$^{Giglio}$ Oligopaint FISH foci [40] was observed in GSCs indicating no obvious aneuploidy defects (S4L–S4R Fig). These results show that the extent of CENP-C depletion (60% reduction) does not result in a mitotic arrest nor in major chromosome segregation defects at 5 days old, indicating that the canonical kinetochore function of CENP-C is maintained. Moreover, mitotic delay or arrest does not explain the observed cell proliferation phenotype. We then used EdU incorporation to label cells with or without newly replicated DNA as a marker of S-phase (S5A–S5D' Fig). Strikingly, we noted that the percentage of EdU positive GSCs increased from 15% in the *nanos-GAL4* control to 35% in the CENP-C RNAi (S5E Fig). We also noted that GSC-CB, 2-, 4- or 8-cell cysts were more frequently observed in CENP-C depleted germaria (S5F Fig). Quantitation showed that while 0–3 EdU positive cysts (mean of 1.04±0.05) were observed in *nanos-GAL4* germaria, this number increased in CENP-C RNAi (mean of 1.68±0.07) (S5F Fig). This suggests that GSCs and cysts in the CENP-C RNAi are either blocked or progress slower through DNA replication in S-phase, perhaps contributing to the observed accumulation of germ cells in germaria.

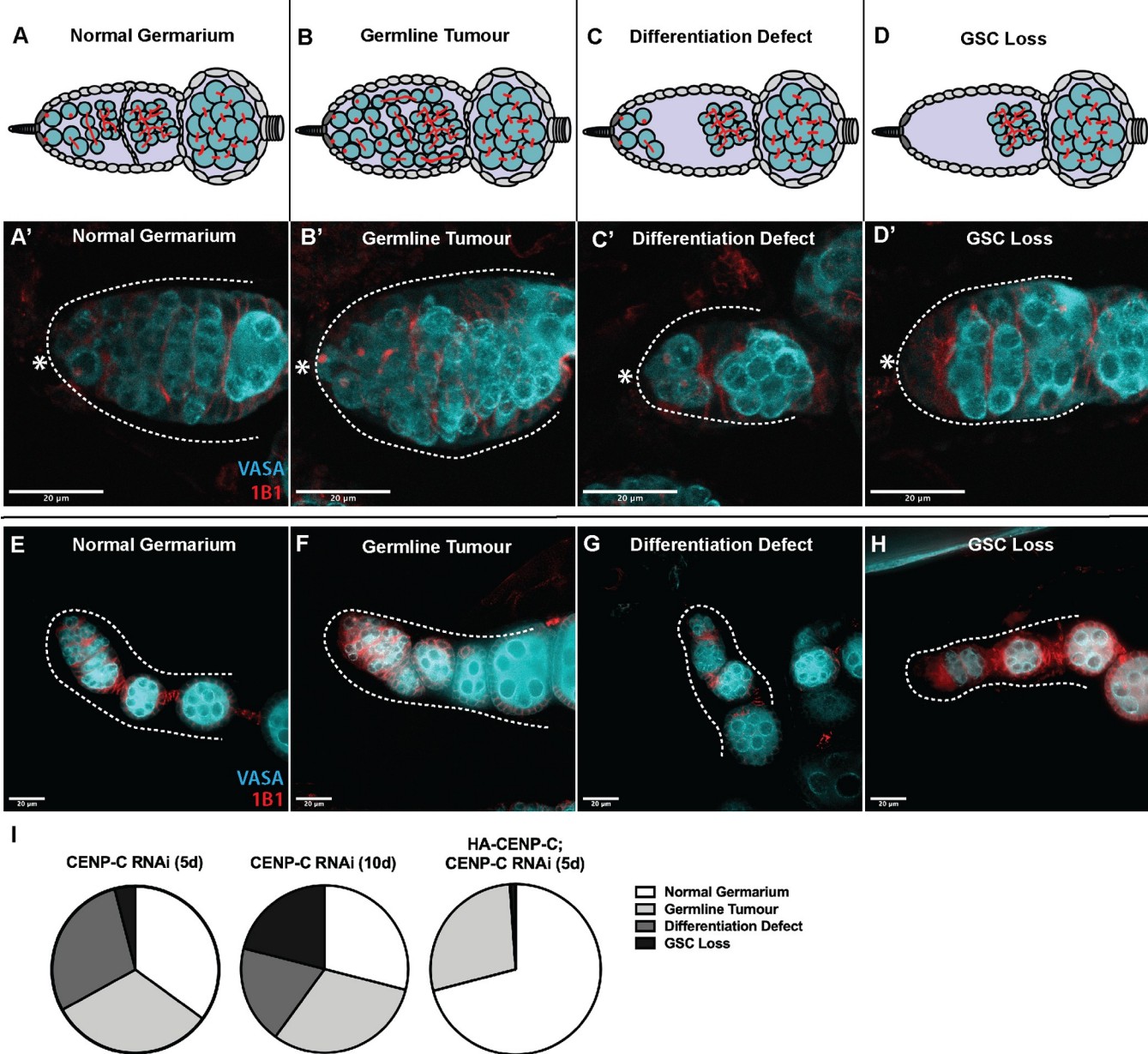

**Fig 4. CENP-C depletion disrupts GSC proliferation and maintenance over time.** (A-D) Characterisation of the phenotypes arising in CENP-C depleted germaria. (A, A') Normal germarium are healthy with the expected lineage of germ cysts and spectrosome/fusome development. (B, B') Germline tumours are characterised by an increased number of germ cells (GSCs, CBs, cysts) in the germaria, often displaced from their normal position with abnormal spectrosome/fusome morphology. (C, C') The differentiation defect is characterised by a pool of GSCs/CBs in the apical end of the germaria, separated from later stage developing cysts. (D, D') GSC loss is characterised by the absence of GSCs (and often CBs and early germ cysts) at the apical end of the germarium. * denotes cap cells. Scale bar = 20 μm. Continued egg chamber development in CENP-C RNAi ovarioles displaying normal germaria (E), germline tumours (F), differentiation defects (G) or GSC loss (H). Scale bar = 20 μm. (I) Quantitation of the frequency of the above phenotypes observed in germaria at 5-days (5d) and 10-days (10d) post-eclosion, and in the HA-CENP-C; CENP-C RNAi rescue at 5-days (5d) post-eclosion. Charts each represent 3 biological replicates (50 germaria analysed per replicate).

## CID and CENP-C levels are reduced in aged GSCs and CENP-C reduction accelerates CID loss

*Wild type* GSCs retain 1.2-fold more CID in an asymmetric division. However, given that symmetric GSC divisions (in which the CID ratio is presumably 1.0) also occur [41,42], we hypothesised that CID and CENP-C levels would gradually change in GSCs over time (Fig 5). We investigated this possibility in *wild type OregonR* GSCs dissecting at 5-, 10- and 20-days post-eclosion, staining for 1B1 to mark GSCs in $G_2$/prophase and either CID (Fig 5A–5C) or CENP-C (Fig 5E–5G). Quantitations showed a significant decrease in CID level between 5- and 20-day timepoints ($OregonR_{5\text{-day}}$ = 0.27±0.03, n = 24 cells; $OregonR_{10\text{-day}}$ = 0.20±0.01, n = 29 cells; $OregonR_{20\text{-day}}$ = 0.17±0.01, n = 26 cells) (Fig 5D). Similarly, CENP-C significantly decreased from 5- and 20-day timepoints ($OregonR_{5\text{-day}}$ = 0.25±0.03, n = 26 cells; $OregonR_{10\text{-day}}$ = 0.19±0.02, n = 28 cells; $OregonR_{20\text{-day}}$ = 0.15±0.01, n = 29 cells) (Fig 5H). Hence, CID and CENP-C levels in GSCs reduce in correlation with GSC age. We next wanted to determine if this observed reduction in CID was dependent on CENP-C. For this, we quantified CID in 5- and 10-day old germaria in both *nanos-GAL4* and CENP-C RNAi GSCs at $G_2$/prophase (Fig 5I–5L). In the CENP-C RNAi, we quantified germaria displaying normal and germline tumour phenotypes at 5-days old and the differentiation defect phenotype at 10-days old. In *nanos-GAL4* GSCs controls, we observed a reduction in total CID signal between 5- and 10-days old ($nanos\text{-}GAL4_{5\text{-day}}$ = 0.28±0.02, n = 31 cells; $nanos\text{-}GAL4_{10\text{-day}}$ = 0.22±0.01, n = 33 cells) (Fig 5M), comparable with *wild type* observations. In CENP-C-depleted GSCs at 5- and 10-days old, we found that CID was reduced further (CENP-C RNAi$_{5\text{-day}}$ = 0.18±0.01, n = 30 cells; CENP-C RNAi$_{10\text{-day}}$ = 0.13±0.01, n = 30 cells). These results support a role for CENP-C in long-term CID maintenance in aged GSCs.

## CENP-C regulates the balance of GSCs and CBs in the niche

To specifically explore CENP-C function in GSC maintenance, we assayed the GSC/CB balance in CENP-C depleted germaria. To measure GSC/CB balance, we used the stem cell marker pMad [43] and SEX-LETHAL (SXL) that labels the GSC-CB transition up to the 2-cell cyst (2cc) stage [44,45] (Figs 6A–6H" and S6A–S6D"). Firstly, in *OregonR* (*wild-type*) and RNAi isogenic control lines, we counted the number of pMad-positive and SXL-positive cells in each germaria at 5-days (S6E Fig). We next used this data to calculate the SXL/pMad ratio as a measure for the number of GSCs compared to CBs and 2ccs in each germarium (S6F Fig). In both controls, although the number of positive pMad and SXL cells differ (S6E Fig), the SXL/pMad ratio remained similar, with approximately 4 SXL-positive cells for every 1 pMad-positive cell at 5-days old (S6F Fig). Analysis of *OregonR* germaria at 10- and 20-days old revealed an unexpected gradual decrease in the SXL/pMad ratio ($OregonR_{10\text{day}}$3.55 ± 0.16; $OregonR_{20\text{day}}$3.12 ± 0.12) and thus a change in the balance in stem/daughter cells over time (S6F Fig). In *nanos-GAL4* 5-day old germaria, we counted approximately 1.5 pMad-positive cells and 6 SXL-positive cells on average (Fig 6I). Therefore, *nanos-GAL4* controls have 4 SXL-positive cells for each pMad-positive cell at 5-days post eclosion (4.15 ± 0.21) (Fig 6J). At 10 days, this ratio dropped (3.69 ± 0.21) (Fig 6J), albeit not significantly. In the CENP-C RNAi germaria analysed at 5-days post-eclosion, the number of pMad-positive cells increased to 2.5 on average, while the number of SXL-positive cells did not change (Fig 6I). As a result, the SXL/pMad ratio is reduced to 2.7:1 (2.68 ± 0.17) (Fig 6J). This ratio for CENP-C RNAi is further reduced to 2.0:1 at 10-days post eclosion (1.99 ± 0.14) (Fig 6J). In contrast, overexpression of HA-CENP-C alone did not change the SXL/pMad ratio. In this case, HA-CENP-C expressing germaria dissected at 5-days old showed approximately 2 pMad positive cells on average, but approximately 8 SXL positive cells (Fig 6I). Thus, the SXL/pMad ratio remained normal at

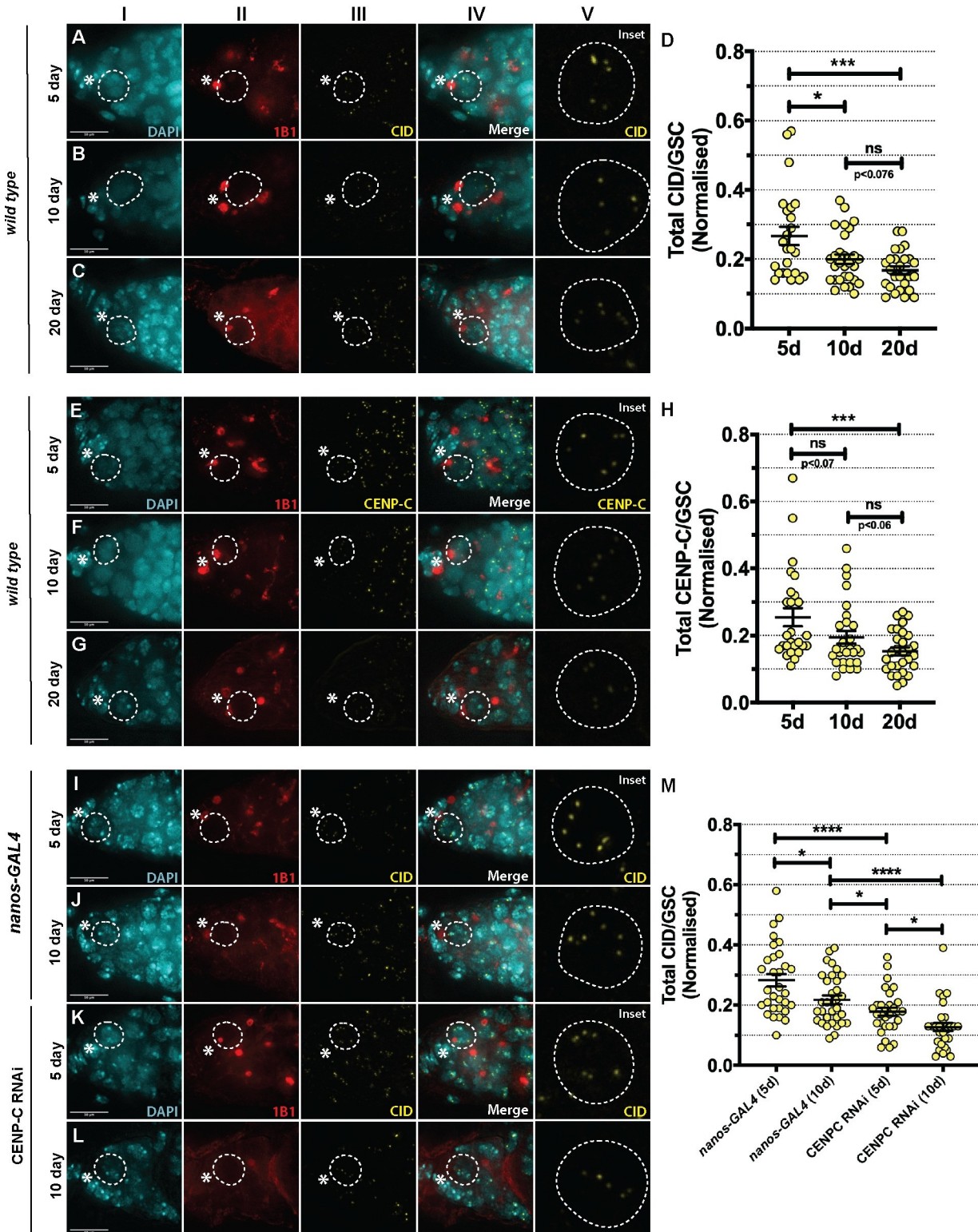

**Fig 5. CID and CENP-C level is reduced in aged GSCs and CENP-C depletion accelerates CID loss.** (A-C) Wild type germaria (5-, 10- and 20-day old) stained with DAPI (cyan), 1B1 (red) and CID (yellow) or (E-G) CENP-C (yellow). GSCs are boxed and inset. * denotes cap cells. Scale bar = 10 μm. Quantitation of total CID (D) or CENP-C (H) integrated density in wild type GSCs at 5-, 10- and 20-days post eclosion. *p<0.05, ***p<0.001, ns = non-significant. Error bars = SEM. (I-L) Germaria of nanos-GAL4 (5d, 10d) and CENP-C RNAi (5d, 10d differentiation defect phenotype) stained with DAPI (cyan), 1B1 (red) and CID (yellow). GSCs are boxed and inset. * denotes cap cells. Scale

bar = 10 $\mu$m. (M) Quantitation of total CID integrated density per GSC in nanos-GAL4 (5d, 10d), CENP-C RNAi (5d, 10d differentiation defect phenotype). *p<0.05, ****p<0.0001. Error bars = SEM.

approximately 4 (3.78 ± 0.19) (Fig 6J). Significantly however, HA-CENP-C overexpression was sufficient to almost fully rescue the disrupted SXL/pMad ratio observed in the CENP-C RNAi (3.58 ± 0.12) (Fig 6J). Finally, CENP-C RNAi carried out in adults using the *nanos-GAL4*; *tubulin-GAL80ts* driver also significantly reduced the SXL/pMad ratio compared to the control in 15-day old flies (*nanos-GAL4; tubGAL80ts* = 3.51±0.15; CENP-C RNAi; *tubGAL80ts* = 2.59±0.08) (Fig 6G–6H", 6I and 6J). Taken together, these results indicate that (1) the balance of stem/daughter cells in the niche slowly changes with age (after 20 days, S6F Fig), and (2) GSCs with reduced CENP-C shift the balance of stem/daughter cells toward self-renewal rather than differentiation, offering an explanation for the differentiation or cell cycle defects we see at 5- and 10-days old (Fig 4).

## Discussion

### CENP-C contributes towards mitotic drive by facilitating CID assembly, maintaining CID asymmetry and assembling a strong GSC kinetochore

Drosophila GSCs use the strength differential between centromeres to bias sister chromatid segregation between stem and daughter cells [20,21]. This asymmetry in centromere strength is achieved through differential CID assembly in $G_2$/prophase, which is used to build a stronger kinetochore and mitotic spindle [20,21]. We have previously shown that CENP-C is asymmetrically distributed between GSC-CB pairs after cell division [20]. We now show that similar to CID, CENP-C is also assembled in $G_2$/prophase of the cell cycle. Moreover, we find that CENP-C is required for CID assembly at this cell cycle time. This direct role for CENP-C in CID (CENP-A) assembly has been previously characterised, mostly in cultured cells [25,28,46]. However, few studies have investigated aberrant centromere assembly in stem cells or in the context of tissue development. Here, we show that defective CID/CENP-C assembly has a profound effect on GSC maintenance and in turn oocyte development over time. In addition to its function in assembly, we find that CENP-C is required to maintain the correct level of CID asymmetry between stem and daughter cells. Specifically, depletion of CENP-C gives rise to GSCs retaining 1.44-fold more CID compared to 1.2 in the controls. Given that CENP-C over-expression was not sufficient to drive CID asymmetry, we suggest that CENP-C's function in asymmetry is likely due to its canonical role in CID assembly. CENP-C might function differentially to maintain CID in GSCs and CBs. It is also possible that CENP-C functions directly in establishing CID asymmetry. In contrast to CENP-C over-expression, CAL1 overexpression (together with CID) in GSCs resulted in a CID ratio of 1 [20], suggesting different functions for CAL1 and CENP-C. In any case, it appears that distorting CID asymmetry (to either 1.0 or 1.4) correlates with a disrupted balance of stem and daughter cells in the ovary. How CENP-C functions together with CAL1 to maintain the correct level of asymmetry remains unclear, however it may relate to different requirements for CAL1 and CENP-C in maintaining pools of newly synthesized or parental CID at distinct cell cycle times. Ultimately, our findings for CENP-C function in GSCs are in agreement with the mitotic drive model for stem cell regulation [23].

### How might parental CID be maintained in stem cells?

Previous studies in Drosophila male GSCs and intestinal stem cells (ISCs) have shown that parental CID, as opposed to newly synthesized CID, is preferentially maintained by stem cells

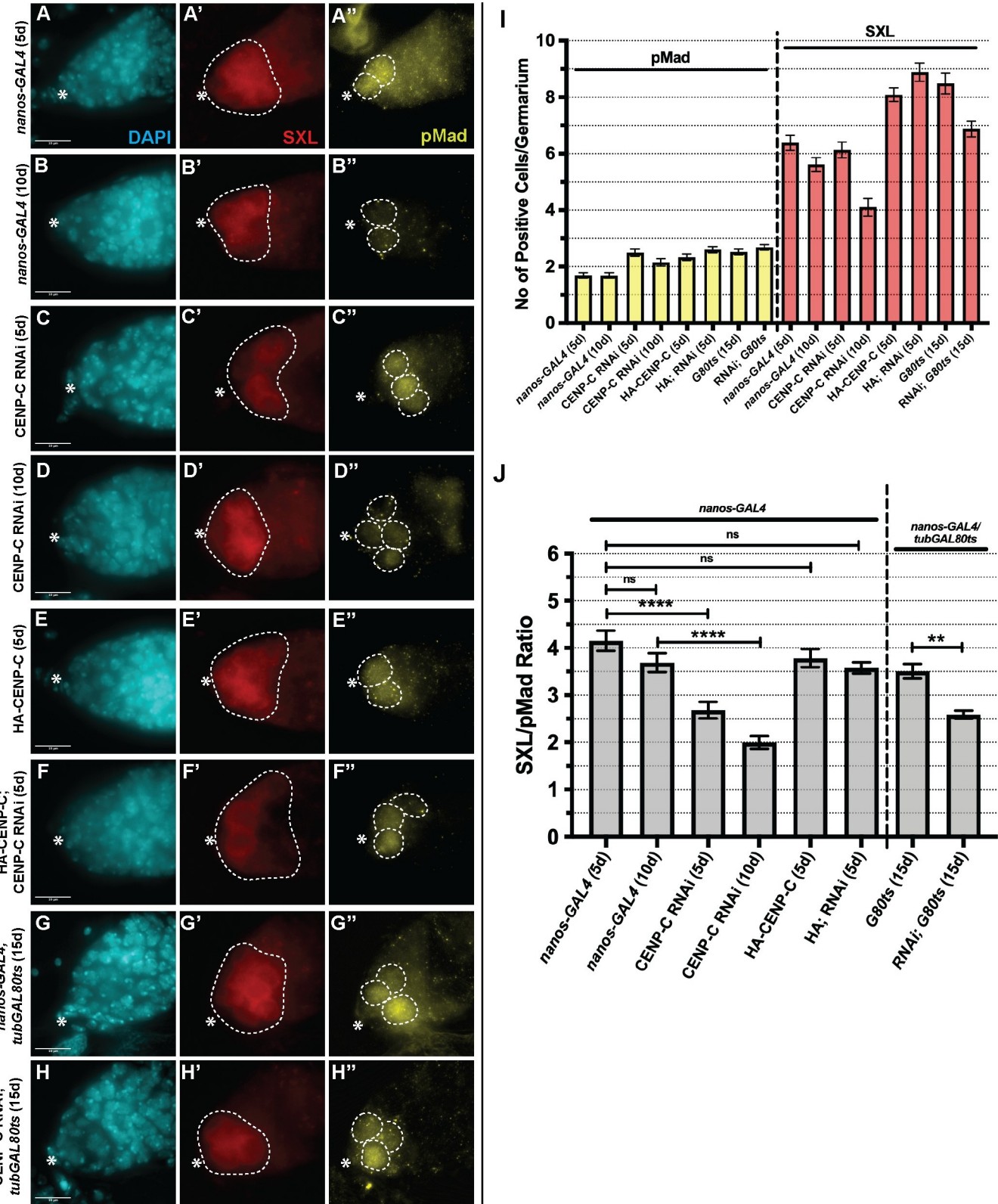

**Fig 6. CENP-C depletion shifts GSCs toward a self-renewal tendency.** (A-F) nanos-GAL4 (5d, 10d), CENP-C RNAi (5d, 10d), HA-CENP-C (5d) and HA-CENPC;CENPC RNAi rescue (5d) germaria and (G) nanos-GAL4; tubGAL80ts (15d), (H) CENP-C RNAi; nanos-GAL4; tubGAL80ts (15d) germaria stained with DAPI (cyan), SEX-LETHAL (SXL, red) and pMad (yellow). Scale bar = 10 $\mu$m. * denotes cap cells. White dashed circles highlight SXL or pMad

positive cells. Images are projections of z-stacks that capture total pMad/SXL signal per germarium. (I) Quantitation of the number of pMad positive (yellow) and SXL positive (red) cells per germarium (n = 40–45). (J) Ratio of the number of SXL:pMad positive cells per germarium. **** p<0.0001, ** p<0.01. ns = non-significant. Error bars = SEM.

[21,22]. Given the lack of centromere assembly in CENP-C-depleted GSCs in our case, we can deduce that these GSCs contain largely parental CID. Furthermore, our results suggest a bias in the retention of parental CID by the stem cell. Many questions surround how parental CID could be maintained at centromeres. In the male germline, testes-derived DNA and chromatin fibres display a high frequency of unidirectional fork movement [9], providing a potential mechanism as to how old versus new histone asymmetry might be established and maintained. The timing for CID assembly in $G_2$/prophase (after DNA replication and sister centromere establishment) indicates that parental CID is redistributed at the replication fork before new CID assembly occurs. It is therefore likely that parental CID requires direct maintenance via CENP-C, CAL1 or other histone chaperones. Intriguingly, in CENP-C-depleted germaria, we frequently observed germ cell cysts (GSC, CB, 2cc, 4cc, 8cc) in S-phase (S5 Fig), suggesting a potential non-canonical function for CENP-C at this cell cycle time. Indeed, previous photo-bleaching experiments in human cell lines showed that unique from most other centromere proteins CENP-C is stable during S-phase [47]. More recently, CENP-C has been shown to maintain centromeric CENP-A in S-phase and allow for error-correction of CENP-A assembly at non-centromere sites [48]. Furthermore, HJURP (functional CAL1 equivalent in humans) is required to maintain CENP-A during DNA replication [49]. It is tempting to speculate that in addition to canonical functions in centromere assembly, CENP-C and/or CAL1 might be utilised in S-phase to establish or maintain CENP-A asymmetry in stem cells.

## Adult stem cells age epigenetically at the centromere

Numerous studies have shown that the epigenome changes with age ('epigenetic drift'), particularly related to DNA methylation, histone modifications and chromatin remodeling [50,51]. Importantly, this epigenetic 'erosion' also pertains to stem cells [52,53]. In this context, an epigenetic regulator of aging should ideally decrease over time and directly influence cell fate. Here we show that both CID and CENP-C decrease approximately 40% on average between 5- and 20-days old in *wild type* GSCs. This loss is further exacerbated upon a reduced CENP-C level, suggesting that CENP-C is directly involved in this centromere 'erosion'. It is likely that the low frequency of symmetric stem cell divisions [41,42] (and in turn symmetric CID distribution) gradually depletes these centromere proteins over time. To our knowledge, this is the first time that the centromere has been implicated in stem cell aging and is consistent with an early study showing centromere loss in aged women [54].

## Centromeres as regulators of stem cell fate and differentiation

By measuring the ratio of stem to daughter cells, we show firstly that the balance of stem to daughter cells in the niche changes gradually over time. Secondly, disruption to the centromeric core by depletion of CENP-C shifts the balance towards GSC self-renewal (reducing the SXL/pMad ratio), and this is exacerbated further over time. At this point we cannot exclude the possibility that these GSCs might result from dedifferentiation, which can occur in Drosophila germaria [55]. Later in development, we observe differentiation defects, measured by an absence of germ cell cysts in the germarium. Thus, CID levels are closely linked with stem cell self-renewal rate, which in turn manifests in a failure in differentiation at cyst-stages, followed by GSC loss. Indeed, CENP-C has been previously implicated in Drosophila stem cell maintenance and/or differentiation, being 1 of 42 genes identified in three different RNAi

screens [36,56,57]. Recently, it has been shown that reprogramming human fibroblasts to pluripotency results in a removal of CENP-A from the centromere [58]. Moreover, low levels of CENP-A prevent human pluripotent stem cells from differentiating, resulting in continuous self-renewal [59]. This implies a certain centromere 'load' required to differentiate–a prospect reinforced by our observations in the germline. Thus, we propose a two-fold role for the centromere in cell fate, wherein 1) centromere 'load' and 2) parental CID/CENP-A pools being key regulators in stem cell fate. How CENP-A load ultimately leads to a change in gene expression should be a focus of future studies.

## Materials and methods

### Fly stocks and husbandry

Stocks were cultured on standard cornmeal medium (NUTRI-fly) preserved with 0.5% propionic acid and 0.1% Tegosept at 20˚C under a 12 hour light-dark cycle. All fly stocks used were obtained from Bloomington Stock Centre (#) unless otherwise stated. The following fly stocks were used: *Oregon-R* (#2371), *wild-type* (#36303, RNAi isogenic control), *UAS-dcr2; nanos-GAL4* (#25751), *nanos-GAL4; tub-GAL80ts* (kind gift from Yukiko Yamashita), *bam-GAL4* (kind gift from Margaret T. Fuller), UAS-CENP-C RNAi (#38917), UASp-HA-CENP-C; *SM6 Cy* (kind gift from Kim S. McKim), HA-CENP-C; UAS-CENP-C-RNAi (this study). CENP-C knockdown (and rescue) using the *nanos-GAL4* driver was performed at 22 $^{o}$C and using the *bam-Gal4* driver at 29 $^{o}$C. For CENP-C knockdown using the *nanos-GAL4; tub-GAL80ts* crosses were set at 20˚C and progeny were shifted to 29 $^{o}$C upon eclosion. HA-CENP-C was induced using *nanos-GAL4* at either 25 $^{o}$C or at 22 $^{o}$C for rescue experiments. $F_1$ progeny were dissected 5, 10, 15 or 20 days after eclosion. Results obtained from each experiment rely on three biological replicates, unless otherwise specified.

### Immunofluorescence (IF)

After fixation, samples were immediately washed in 1XPBS-0.4% Triton-X100 (0.4% PBST). Samples were then blocked in 0.4% PBST with 1% BSA for 2–4 hours at room temperature, incubated with primary antibodies (in blocking buffer) overnight at 4˚C. Samples were then washed in 0.4% PBST for 3x 30 minutes. Secondary antibodies are added (1:500 in blocking buffer) for 2 hours at room temperature in the dark. Samples are again washed 3x 30 minutes in 0.4% PBST followed by addition of DAPI (1:1000) for 15 minutes in 1XPBS.

### EdU Incorporation

Ovaries were dissected and incubated for 30 min with EdU (0.01 mM) in 1XPBS and then fixed as described. After washing in 0.4% PBST, ovaries were incubated for 30 minutes in the dark with 2 mM CuSO$_4$, 300 μM fluorescent azide and 10 mM ascorbic acid. Samples were then washed with 0.4% PBST for 10 minutes and then blocked and stained as described above.

### Oligopaint IF-FISH

Oligopaint probe for Cen3$^{Giglio}$ was synthesized from Oligopaint library (gift from Barbara Mellone) according to protocol previously published [40]. Ovaries were dissected in 1X PBS (8–10 flies per prep) and ovarioles teased apart, followed by fixation in 200 μl 4% PFA/1XPBS/ 0.5% NP-40 plus 600 μl. Samples were shaken vigorously by hand (should turn milky white) and placed on a rotator, washed 3 X 5 minutes in 1X PBS + 0.1% Tween-20 (hereafter PBT) and blocked for 2 hours in 1.5% BSA in PBT. Primary antibodies were added overnight at 4˚C. The following day, samples were washed 3X 20 minutes in PBT followed by incubation with

secondary antibodies in 1.5% BSA for 2 hours at room temperature. Samples were then washed 2X 20 minutes in PBT followed by 20 minutes in 1X PBS. Samples were washed quickly 3X in 2XSSCT, followed by 1X 10 minute wash in 2XSSCT + 20% Formamide, and 1X 10 minute in 2XSSCT + 50% Formamide. Samples were then washed in 2XSSCT + 50% Formamide at 37˚C for 4 hours on a shaker. Cen3$^{Giglio}$ probe was added (20 pmol) in 2XSSCT/10% dextran sulfate/0.1% Tween-20/50% Formamide + 1μl RNAse A (40 μl total reaction, in PCR tube). Samples were denatured for 30 minutes at 90˚C in a thermocycler followed by hybridisation overnight at 37˚C. The following day, samples were washed 2X 30 mins in 2XSSCT + 50% Formamide at 37˚C on a shaker. 40 pmol of Alexa Fluor 488 secondary probe [40] was added in hybridisation solution (40 μl reaction) at 37˚C in a thermocycler. Samples were washed twice (30 minutes each) in 2XSSCT + 50% Formamide at 37˚C followed by once in 2XSSCT + 20% Formamide for 10 minutes at room temperature. Samples were rinsed 4X quickly in 2XSSCT and moved into 1XPBS. Hoechst was added at 1:1000 for 10 minutes followed by one wash in PBT, and then mounted on a slide in SlowFade mounting media.

## Antibodies

For immunostaining, the following antibodies were used: rabbit anti-CENP-A (CID) antibody (Active Motif 39719; 1:1000), rat anti-CID antibody (Active Motif 61735, 1:500), sheep anti-CENP-C (Dattoli *et al*, 2020; 1:2000), mouse anti-H3S10P (Abcam ab14955; 1:1,000), rabbit anti-VASA (Santa Cruz sc-30210; 1:300), rat anti-VASA (Developmental Studies Hybridoma Bank (DSHB); 1.500), mouse anti-Hts (1B1, DSHB; 1:500), rabbit anti-CAL1 (Bade *et al*, 2014; 1:1000), rabbit anti-SMAD3/5 (pMad) (Abcam; 1:500), mouse anti-SEX-LETHAL (DSHB, M114, 1:500), sheep anti-Spc105 (M. Przewloka; 1:2000), DAPI (1:1000), Hoechst (1:1000).

## Widefield microscopy

Images of immunostained ovaries mounted in SlowFade Gold antifade reagent (Invitrogen S36936) were acquired using a DeltaVision Elite microscope system (Applied Precision) equipped with a 100x oil immersion UPlanS-Apo objective (NA 1.4). Images were acquired as z-stacks with a step size of 0.5 μm. Fluorescence passed through a 435/48 nm; 525/48 nm; 597/45 nm; 632/34 nm band-pass filter for detection of respectively DAPI, Alexa Fluor 488, mCherry and Alexa Fluor 647 in sequential mode.

## Confocal microscopy

Images for Fig 4A'–4D' were taken using an inverted Fluoview 1000 laser scanning microscope (Olympus) equipped with a 60× oil-immersion UPlanS-Apo objective (NA 1.2). The samples were excited at 404, 473, 559, and 635 nm, respectively, for DAPI and Alexa Fluor 488, 546, and 647. Light was guided to the sample via D405/473/559/635 dichroic mirror (Chroma). The pinhole was set at 115 μm. Fluorescence was passed sequentially through a 430–455-, 490–540-, 575–620-, 655–755-nm bandpass filter for detection of DAPI and Alexa Fluor 488, 546, and 647. Images were acquired as z-stacks with a step size of 0.5 μm.

## Quantification

For each quantification one cell/germarium was considered. Images from a single cell (nucleus) were projected (max intensity) to capture all the centromeres present in the cell at a specific cell cycle phase. Image J software [60] was used to measure fluorescent intensity of CID in the following way: The background was subtracted from the projected image. Threshold was adjusted and the image. Size was adjusted, in order to eliminate unwanted objects.

Following, the command "analyse particles" was used to select centromeres. Finally, integrated density (MGV*area) from each centromere foci were extracted and used as fluorescent intensity to measure the total amount of fluorescence per nucleus. Quantification of pMad and SEX-LETHAL positive cells was obtained by scanning the z-stack of each image to count cells with specific signals and to distinguish from any background signals. Quantitations in Fig 5 were normalised to spectrosome fluorescence in each respective germarium. After z-projection, a 1 μm x 1 μm box was drawn inside the GSC spectrosome fluorescence and Integrated Density was measured. This value was divided into the CID or CENP-C value calculated for each respective GSC.

### Statistical analyses

Data distribution was assumed to be normal, but this was not formally tested. P value in each graph shown was calculated with unpaired t test or One-way Analysis of Variance (ANOVA) with tukey's test for Fig 6J. All statistical analysis was performed using Prism 9 software.

### Supporting information

**S1 Fig. Characterisation of CENP-C level in control *nanos-GAL4*, CENP-C RNAi, HA-CENP-C and HA-CENP-C; CENP-C RNAi lines.** Immunofluorescent image of 5-day old (5d) $G_2$/prophase GSCs (circled) in (A-D) *nanos-GAL4* control, (A'-D') CENP-C RNAi and (A"-D") HA-CENP-C; CENP-C RNAi stained with DAPI (cyan), CENP-C (green) and 1B1 (red). Scale bar = 5 μm. CENP-C RNAi and rescue experiments were performed at 22˚C. (E) Quantitation of total CENP-C fluorescent intensity (integrated density) per GSC in *nanos-GAL4*, CENP-C RNAi and HA-CENP-C; CENP-C RNAi (rescue). ****p<0.0001, ns = non-significant. Error bars = SEM. (F-I) Immunofluorescence image of 5-day old (5d) $G_2$/prophase GSCs (circled) over-expressing HA-CENP-C stained with DAPI (cyan), CENP-C (green) and 1B1 (red). * denotes cap cells/GSC niche. GSCs are circled. Scale bar = 5 μm. HA-CENP-A over-expression experiments were performed at 25˚C. (J) Quantitation of total CENP-C fluorescent intensity (integrated density) per GSC in HA-CENP-C. ***p<0.001. Error bars = SEM. (TIF)

**S2 Fig. CENP-C is required for CAL1 localisation in GSCs, but it is not required for CID localisation at later stages of development.** Immunofluorescent image of $G_2$/prophase GSCs (circled) in (A-C) *nanos-GAL4* and (A'-C') CENP-C RNAi stained for DAPI (cyan), 1B1 (red), CAL1 (yellow) and CENP-C (green). Centromeric CAL1 was identified as being colocalised with CENP-C. GSCs are circled. *denotes cap cells. Scale bar = 5 μm. (D) Quantitation of total centromeric CAL1 fluorescent intensity (integrated density) in *nanos-GAL4* and CENP-C RNAi. ***p<0.001. Error bars = SEM. (E-H) *bam-GAL4* and (E'-H') *bam-GAL4* driven CENP-C RNAi stained with DAPI (cyan), CENP-C (yellow) and 1B1 (red). Circle marks region where knockdown begins. (I-L) *bam-GAL4* and (I'-L') *bam-GAL4* driven CENP-C RNAi stained with DAPI (cyan), H3S10P (red) to mark 8-cell cysts in mitosis (circled) and CENP-C (yellow). (M) Quantitation of CENP-C in each cell of 8-cell cysts of *bam-GAL4* and CENP-C RNAi. **p<0.01. Error bars = SEM. (N-Q) *bam-GAL4* and (N'-Q') *bam-GAL4* driven CENP-C RNAi stained with DAPI (cyan), H3S10P (red) to mark 8-cell cysts in mitosis (circled) and CID (yellow). (R) Quantitation of CID in each cell of 8-cell cysts of *bam-GAL4* and CENP-C RNAi. ns = non-significant. Error bars = SEM. * denotes cap cells. Scale bar = 10 μm. (TIF)

**S3 Fig. Quantitation of CID level in GSCs and CBs at S-phase.** Quantitation of total CID fluorescent intensity (integrated density) in S-phase GSCs and CBs in nanos-GAL4 and (A)

CENP-C RNAi or (B) HA-CENP-C or (C) HA-CENP-C; CENP-C RNAi (rescue). Each point represents the total CID integrated density per GSC/CB nucleus. **p<0.01. ns = non-significant. Error bars = SEM. (D-G, H-K) nanos-GAL4; tub-GAL80ts and (D'-G', H'-K') nanos-GAL4; tub-GAL80ts driven CENP-C RNAi stained with VASA (cyan), 1B1 (red) and CENP-C (yellow). Progeny were analysed at 5 (5d, D-G') and 15 days (15d, H-K') post eclosion. White circle outlines CENP-C depleted regions. * denotes cap cells. Scale bar = 20 $\mu$m.
(TIF)

**S4 Fig. GSCs and germaria with reduced CENP-C do not exhibit obvious defects in mitosis.** (A-D) 5 day old *nanos-GAL4* and (A'-D') CENP-C RNAi (germline tumour phenotype) stained with DAPI (cyan), VASA (grey) and H3S10P (red). * denotes cap cells. Scale bar = 10 μm. (E) Mitotic index (%) of H3S10P positive GSCs in *nanos-GAL4* and CENP-C RNAi (n = 150 germaria). (F) Violin plot displaying the number of H3S10P positive cysts per germaria (n = 150 germaria). One positive hit was quantified as H3S10P positive GSC-CB pairs, 2-cell cysts (2cc), 4-cell cysts (4cc) or 8-cell cysts (8cc). ns = non-significant. (G-K) 1 day old *nanos-GAL4* and (G'-K') CENP-C RNAi stained with DAPI (cyan), histone H3 phosphorylated on threonine 3 (H3T3P) to mark prometaphase GCSs (blue), Spc105 (yellow) and 1B1 (red). 1 day old flies were analysed for this experiment in order to isolate prometaphase in actively dividing GSCs (circled). * denotes cap cells. Scale bar = 5 μm. (L-Q) *nanos-GAL4* and (L'-Q') CENP-C RNAi stained with Hoescht (cyan), 1B1 (red), CID (grey) and Cen3$^{Giglio}$ oligopaint FISH (yellow). White arrows indicate Cen3 that overlap with CID foci. GSCs are circled. (R) Quantitation of CID foci per GSC (grey bars) or Cen3 foci (overlapping with CID) per GSC (yellow bars) in *nanos-GAL4* and CENP-C RNAi (n = 30 GSCs). ns = non-significant. Error bars = SEM.
(TIF)

**S5 Fig. Germaria with reduced CENP-C exhibit a higher frequency of S-phase cells.** (A-D) 5 day old *nanos-GAL4* and (A'-D') CENP-C RNAi stained with DAPI (cyan), EdU (yellow) and 1B1 (red). * denotes cap cells. White dashed lines outline EdU positive GSC-CB (top) or cysts (bottom). Scale bar = 10 μm. (E) S phase index (%) of EdU positive GSCs in *nanos-GAL4* and CENP-C RNAi (n = 150 germaria). (F) Violin plot showing the quantitation of EdU positive cysts per germarium (n = 150 germaria). One positive hit was quantified as a single EdU positive GSC/CB, 2-cell cysts (2cc), 4-cell cysts (4cc) or 8-cell cysts (8cc). ****p<0.0001.
(TIF)

**S6 Fig. A method of measuring female GSC self-renewal versus differentiation.** (A-C) *Wild type* (*OregonR*) (5-, 10- and 20-days old) and (D) *wild type* (#36303, RNAi isogenic control) germaria stained with DAPI (cyan), SXL (red) and pMad (yellow). *denotes cap cells. Scale bar = 10 μm. White dashed circles highlight SXL or pMad positive cells. (E) Quantitation of the number of pMad positive (left, yellow) and SXL positive (right, red) per germarium (n = 40–45). (F) Ratio of the number of SXL:pMad positive cells per germarium. ***p<0.001, *p<0.05, ns = non-significant. Error bars = SEM.
(TIF)

**S1 Data. Numerical data underlying graphs and summary statistics.** Data tables for Figs 1–6 and S1–S6 are provided in.xml format.
(XML)

## Acknowledgments

The authors acknowledge the facilities and technical assistance of the Centre for Microscopy & Imaging at the National University of Ireland Galway (www.imaging.nuigalway.ie). Stocks

were obtained from the Bloomington Drosophila Stock Center (NIH P40OD018537). Antibodies obtained from the Developmental Studies Hybridoma Bank, created by the NICHD of the NIH are maintained at The University of Iowa, Department of Biology, Iowa City, IA 52242. We thank Sylvia Erhardt for rabbit anti-CAL1 antibodies and Marcin Przewloka for anti-Spc105 antibodies. We thank Barbara Mellone for Cen3$^{Giglio}$ Oligopaint library and for advice on IF-Oligopaint protocol in ovaries. We also thank Eric Joyce for advice on IF-Oligopaint protocol. We thank Annie Walshe for generation of the sheep anti-CENP-C aa 1–732 antibody and Kim McKim for the HA-CENP-C fly line.

## Author Contributions

**Conceptualization:** Ben L. Carty, Anna A. Dattoli, Elaine M. Dunleavy.

**Formal analysis:** Ben L. Carty.

**Funding acquisition:** Ben L. Carty, Anna A. Dattoli, Elaine M. Dunleavy.

**Investigation:** Ben L. Carty.

**Methodology:** Ben L. Carty, Anna A. Dattoli, Elaine M. Dunleavy.

**Project administration:** Elaine M. Dunleavy.

**Resources:** Ben L. Carty, Elaine M. Dunleavy.

**Supervision:** Anna A. Dattoli, Elaine M. Dunleavy.

**Visualization:** Ben L. Carty.

**Writing – original draft:** Elaine M. Dunleavy.

**Writing – review & editing:** Ben L. Carty, Anna A. Dattoli, Elaine M. Dunleavy.

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
