## [Decision Letter · Decision Letter 0]

8 Dec 2020

Dear Dr Dunleavy (Hi Elaine!),

Thank you very much for submitting your Research Article entitled 'CENP-C regulates centromere assembly, asymmetry and epigenetic age in Drosophila germline stem cells.' to PLOS Genetics.

The manuscript was fully evaluated at the editorial level and by 3 independent expert peer reviewers. The reviewers appreciated the attention to an important problem in centromere biology and chromosome inheritance and noted the high standards by which the experiments were performed. However, they raised several substantial concerns about the manuscript in its current form, including data visualization and analyses as well as the strength of the conclusions based on the data presented. Based on the reviews, we will not be able to accept this version of the manuscript, but we would be willing to review a much-revised version. We cannot, of course, promise publication at that time.

Should you decide to revise the manuscript for further consideration here, your revisions should address the specific points made by each reviewer. In particular, we ask that you:

1. provide additional details of experiments and data analysis

2. address possible chromosome segregation defects

3. clarify the data; confirm that selected images in figures match conclusions stated in text

4. distinguish, if possible, if loss of GSCs is due to self-renewal versus de-differentiation

5. provide strong evidence that CENP-C is dispensable for centromere assembly in later developmental stages

We will also require a detailed list of your responses to the review comments and a description of the changes you have made in the manuscript.

If you decide to revise the manuscript for further consideration at PLOS Genetics, please aim to resubmit within the next 60 days, unless it will take extra time to address the concerns of the reviewers, in which case we would appreciate an expected resubmission date by email to plosgenetics@plos.org.

[LINK]

We are sorry that we cannot be more positive about your manuscript at this stage. Please do not hesitate to contact us if you have any concerns or questions.

Yours sincerely,

Beth A. Sullivan, PhD

Associate Editor

PLOS Genetics

John Greally

Section Editor: Epigenetics

PLOS Genetics

Reviewer's Responses to Questions

**Comments to the Authors:**

Reviewer #1: The manuscript entitled ‘CENP-C regulates centromere assembly, asymmetry and epigenetic age in Drosophila germline stem cells’ by Ben Carty et al. deals with the question how Drosophila CENP-A/CID is loaded and maintained in female germ line stem cells. Like in other cells types of Drosophila melanogaster, the authors show that CENP-C is essential for CENP-A localization to the germ line stem cells and its assembly at G2 phase of the cell cycle. In addition, CENP-C is also required for the asymmetric distribution of CENP-A. The authors find that in aging germ line stem cells with reduced CENP-C, CENP-A decline is more rapid. Last but not least CENP-C also seems to be involved in regulating the asymmetric division of germ line stem cells with a higher percentage of symmetric division and, therefore, self-renewal of germ line stem cells.

The manuscript is well written and addresses interesting aspects of centromere biology and reproductive biology also in the context of aging. I would like to raise several points that the authors may want to consider for improving their findings:

It isn’t always clear how the authors distinguished cells in S phase and G2 phase. I understand that it is based on the EdU staining and the spectrosome but the EdU is hard to see because of its overlap with the DAPI staining. A separate panel with EdU staining only should be included. In addition, the spectrosome does not always look extended but sometimes more like a round spectrosome (Fig. 1C, 2A). The quantifications look good and I assume that looking at the actual samples makes it much easier to distinguish it but the authors should choose pictures that reflect what they describe in the text. Including other markers (for instance Vasa would also improve the images and could serve as a signal to normalize CENP-C/CENP-A/CAL1 on, see also further down).

The quantification of Figure S2D isn’t clear. The authors write that they quantified the centromeric CAL1 levels in control and CENP-C depleted cells. The centromeres are, however, defined here by CENP-C which is absent in the depletion. How can the authors measure Cal1 at centromere (and distinguish it from the nucleolar pool).

Quantifying the signal of CENP-A in later developmental stages in control and CENP-C-depleted germarium would be good: The authors say that there is no effect on CENP-A, the provided images looks like there is even more CENP-A (S2J’).

In addition, the authors should consider that at later developmental stages the significant amount of remaining CENPC (?, bam-driven depletion has not been quantified, should be done) is CENP-C-depleted cells may be sufficient for those cells to function (or that the RNAi isn’t as effective as in earlier stages).

In Figure 3 again, the EdU staining is hard to see.

Figure 4. The presentation could be improved. The arrow with time suggests that the different phenotypes are derived from each other, which I don’t think they do nor that the authors wanted to claim this. I would remove the arrow. Importantly, the fact that the germ line tumors are not rescued at all by CENP-C rescue experiments indicates an unresolved proliferation defects cause by too much CENP-C or CENP-C expressed at the wrong time of the cell cycle that the authors should address further, at least in the discussion if not experimentally. The claims in the result section need to be toned down somewhat and this phenomenon discussed.

The increased number of EdU positive cells in CENP-C RNAi germ cells where interpreted as ‘going through S-Phase slower’ when CENP-C is depleted. However, could it also be an S-phase bloc that the authors observe?

How can the graph presented in S4Q (error bars!) be highly significant?

In Fig 5, the authors could have made a more convincing claim if they would have also used readily available stem cell markers such as Vasa. Also, co-staining with a marker that does not change would have controlled for experimental variability and would have given the authors a signal to normalize the fluorescence intensity of CENP-C.

Reviewer #2: This work by E. Dunleavy and colleagues investigated whether CENP-C has a role in maintaining the “mitotic drive” to bias sister chromatid segregation, a novel phenomenon recently reported by several labs including the authors’. The authors are addressing an important question to further understand the underlying mechanisms to enhance our understanding of this phenomenon using Drosophila female germline stem cells as the model system. Importantly, they have shown that CENP-C, an inner kinetochore component, is required to maintain a normal asymmetric distribution of CID and GSC normal function. The approaches take advantage of the powerful molecular genetics and cell biology tools, which are in general well executed. However, some experiments will need additional control and some results will need additional analyses.

Major comments:

1. Throughout the manuscript for the RNAi experiments, the control is using the nanos-GAL4 driver only, it is better to use a control of nanos-GAL4 paired with some non-specific RNAi to activate the RNAi pathway, such as UAS-lacZ RNAi or GFP RNAi, etc. However, for some sets of experiments, the rescuing experiment was performed which showed more gene specificity, it may not be necessary to repeat those experiments when the rescue experiment was performed.

2. Again, for the RNAi experiments, the authors argued that the secondary effect should be minimal. But normally to make the knockdown acute, it is better to combine the nanos-Gal4 driver with the tub-Gal80_ts in order to knockdown at a particular stage, such as adulthood, in order to prevent prolonged knocking down and potential secondary defects.

3. In their previous paper, they showed difference of CID between sister centromeres which is very nice and avoid any complication of cell cycle stage difference. In this manuscript, all data are exclusively in post-mitotic GSC-CB pairs. Examination of changes at individual sister centromeres would be very informative.

4. Centromere proteins are known to be critical to maintain normal mitotic progression and cell cycle progression. Therefore, prolong depletion of centromeric proteins could lead to GSCs loss. And this loss could be caused by failure in GSC self-renewal, or de-differentiation process, a phenomenon reported in this system especially during aging. Therefore, it would be interesting to explore whether the change of GSC number upon CENP-C depletion could be due to dedifferentiation defects.

Minor comments:

1. The sentence “We found an increase in CENP-C level between cells in S-phase, compared to cells in G2/prophase.” is confusing, should be revised to: We found an increase of CENP-C level in G2/prophase cells compared to cells in S-phase.

2. For this conclusion “This result indicates that CENP-C is specifically required for CID assembly that occurs between the end of S-phase up to prophase in GSCs.”, it may be a bit overstretch given the temporal resolution the EdU label can inform.

3. Fig. 2J and 2K, the statistical analyses should also be performed at the comparable cell cycle stage between genotypes, in order to support conclusions such as “no significant change” or “partial rescue”. Similarly, in Fig.5M, the statistical analyses should also be performed at the comparable age between genotypes.

4. Page 7, “we assayed whether cells were blocked in mitosis using phosphorylation at serine 10 of histone H3 (H3S10P) to marks cells at late G2-phase to metaphase.” I guess the authors mean M phase instead of metaphase? H3S10P is detectable throughout M phase including anaphase and telophase. Also, this mark was labeled incorrectly in Figure S4, with a pattern more like at the periphery of the chromosomes? Finally, it is said in the text that “H3S10P combined with DAPI staining of DNA allowed us to monitor for chromosome segregation defects, which were also absent (data not shown).” It would be better to add data in the supplement. The mitotic index would inform any potential mitotic defects. For example, there is no mitotic GSCs in Figure S4, which could be due to cell cycle arrest. In addition, GSCs at anaphase and telophase would be the best stage to examine whether there is any segregation defect.

5. For this statement in the text on page 8, “Given that symmetric GSC divisions (in which the CID ratio is presumably 1) occur at a low frequency [37,38], we hypothesised that CID and CENP-C levels would gradually be depleted in GSCs over time (Fig 5).” I do not quite understand the rationale here.

6. The pMAD staining signals in Figure 6A” and S5C”and S5D” have lots of background, which would make it hard to count the exact number of positive cells for this marker.

7. Page 10 in Discussion, “Ultimately, our findings for CENP-C function in GSCs are in full agreement with the mitotic drive model for stem cell regulation [9].” Wrong reference here.

Reviewer #3: Previous work has shown some unusual characteristics of centromere protein loading in germ line cells. Previous work from the Dunleavy lab has shown that centromere protein CENP-A/CID is deposited during G2. In addition, it is partitioned unevenly between centromeres following replication. One possible implication of this is that sister kinetochores do not randomly attach to microtubules, leading to non-random segregation of chromatids and possibly effects on differentiation. This paper focuses on analysis of CENP-C in germ line divisions using two important tools, RNAi to Cenp-C and an RNAi-resistant Cenp-C transgene.

The results are solid with carefully constructed controls. For example, there are separate sets of nos-Gal4 controls in Fig 3. In addition, most of the images are nicely presented, and the centromere signals are easy to see. However, it can be a struggle to see the 20% differences represented in the graphs. For example, the image in 1D’ is apparently brighter than 1C’. Not sure there is a solution to this.

The results are interesting, but many are not surprising based on previous studies on CID recently published in 2020. The results in the first part of this paper show CENP-C behaving much like CID and Cal1. The more interesting results are the germline proliferation defects in CENP-C RNAi. However, there are some concerns because the data hints that there is a defect in S-phase but it is not clear what that is, and the authors do not do a good job of ruling out chromosome segregation defects. These points and additional less significant issues are discussed below.

1) The first set of major findings is that CENP-C shares the unique properties of CID in the germline. It is required for increased CID in G2, and this can be rescued with an RNAi resistant transgene. Overexpression of Cenp-C, however, has little effect on asymmetric CID, suggesting the levels of CENP-C don’t drive this process. Also like CID, CENP-C is asymmetric between the GSC and CB cells. A little surprising result is that CENP-C depletion increases the GSC-CB ratio, possibly indicating that when CENP-C is limiting, preference goes to loading in the GSC.

In this section, the authors should be careful not to overstate the significance of their results. Given the known function of CENP-C to interact with CAL1 and CID in centromere, it is not surprising that, like the previous 2020 publication, CENP-C has a role in G2 loading and asymmetric assembly of CID. Perhaps the authors should discuss whether the role of CENP-C observed is via the known pathway of centromere assembly, or is it modified to result in G2 loading and/or asymmetric inheritance.

More importantly, the authors should be careful not to overstate their results and suggest a direct function of CENP-C in regulating G2 loading and asymmetric inheritance. The heading on pg 6 suggests CENP-C activity promotes asymmetry. The last line of pg 9 makes a similar conclusion. However, observing an effect of CENP-C depletion on G2 loading and asymmetric inheritance is only consistent with a role in centromere assembly. Instead, these sentences imply that CENP-C is responsible for these unique features. To show a direct role of CENP-C, the authors would need to have a separation of function result and show that a specific CENP-C depletion effects only the G2 loading or asymmetric inheritance, while other functions (like building centromeres and kinetochores) are not affected. In fact their results argue the opposite. Overexpression of CENP-C did not affect the asymmetric behavior of CID. Doesn’t this argue against a role of CENP-C in regulating asymmetry? (as opposed to the results with overexpression of CID or CAL1 in the 2020 paper).

2) The more novel results concern the role of Cenp-C in germ cell differentiation RNAi. CENP-C RNAi has a variety of germarium phenotype, and these get more severe with age. This is interesting but can the authors relate this to the biology a little more and discuss why severity increases with age? Nos-Gal4 expression begins in the germline. Does the age effect reflect the time it takes for CENP-C to be depleted? What is the effect on fertility? Are the 5d and/or 10d females fertile? Do they produce embryos? The germ line tumor phenotype is also interesting but not explained. This seems to be the results that most strongly implicates CENP-C in differentiation. Does Cal1 or CID knockdowns have these phenotypes or is this specific to CENP-C?

At the bottom of page 7 the authors conclude that there are no chromosome segregation defects. However, the data is “not shown”. While this result is plausible, given the partial KO of CENP-C and the tumor phenotype, this data needs to be shown since it is not clear how with just H3S10P staining this can be concluded. Instead, the author find that the depletion of CENP-C has an effect on S-phase progression. It would be really interesting if the authors have discovered a new S-phase function for CENP-C in germ line differentiation. However, they have to rule out mitotic defects, which would require more careful analysis such as mitotic index, karyotyping, and staining to markers like kinetochore ands checkpoint proteins. In short, this seems like a missed opportunity to show that partial loss of CENP-C affected differentiation but not cell division.

3) The methods state that RNAi experiments were done at 22deg. This is significant because UAS/GAL4 is typically stronger at higher temps. Why was 25 deg not used? The knockdown may have been stronger and more severe phenotypes observed. Why was 29 used for bam-Gal4 and 25 for HA-CENPC, and what temperature was HA-Cenp-C+RNAi?

Pg 5: 7 lines from bottom: What is the basis for concluding that CENP-C is “dispensable” for later divisions. In the RNAi genotype, CENP-C protein is still visible and only reduced 60% (pg 5). Is the 40% protein localization level because the RNAi is not efficient, or because the protein is very stable. In addition, and for this reason, the evidence does not support the conclusion that Cenp-C is dispensable for centromere assembly in later germarium divisions. The knockdown might be to mild, and bam expression may be too transient to effect CID levels.

4) Figure 6: Is the 10d SXL/pMAD ratio significantly lower than 5d? I am guessing not, and if so, should be stated as such and the conclusion in line 16 can’t be made. In addition, is the shift towards stem cells in 10d females (last line of section) a result of an arrest in cell division (either due to S-phase or mitotic defects).

5) Pg 4, middle – Female GSCs divide… to give a differentiating (CB) and another GSC.

6) Pg 4. It would help to have more description of how you know the EdU-negative cells are in G2 versus G1 (Figure 1). Also, 4 lines from bottom, it might sound better to write that CENP-C levels were higher in G2 than S. The current sentence could be mis-interpreted.

Pg 5, 7 lines from bottom: “occurring in the germarium”

7) Pg 6, line – in addition to reference to Fig 3A, it would help to report the GSC-CB ratio.

8) On pg 10, the authors state: “distorting the CID asymmetry disrupts the balance of stem and daughter cells”. Has this been shown, or is it a correlation? If so, what is the evidence? Similarly, later on pg 10 “It is tempting to speculate that CENP-C …might be utilized to maintain parental CANP-A in a asymmetrically …”. What data supports the idea that CENP-C maintains asymmetry, as opposed to being required for the loading process which is asymmetric.

9) The sentence on pg 10 line 20 sounds interesting , but is not backed up by any data and should be deleted if the data is not shown.

10) Pg 11, line 11, “CID levels”

11) In Figure 3, is HA; RNAi (3G) significantly different than HA without RNAi (3F)?

**Have all data underlying the figures and results presented in the manuscript been provided?**

Reviewer #1: Yes

Reviewer #2: Yes

Reviewer #3: Yes

PLOS authors have the option to publish the peer review history of their article (what does this mean?). If published, this will include your full peer review and any attached files.

Reviewer #1: No

Reviewer #2: No

Reviewer #3: No

---

## [Decision Letter · Decision Letter 1]

13 Apr 2021

Dear Elaine,

Thank you very much for submitting your Research Article entitled 'CENP-C functions in centromere assembly, the maintenance of CENP-A asymmetry and epigenetic age in Drosophila germline stem cells.' to PLOS Genetics. The manuscript was fully evaluated at the editorial level and by the original 3 independent peer reviewers. The reviewers and I really appreciated the changes that you and your team made in response to the previous comments, and in fact, two of the reviewers are completely satisfied. One reviewer (R1) was largely satisfied with the changes, but has two remaining minor concerns that are I believe would benefit from further clarification before the manuscript is accepted for publication.

Specifically, confusion remains about the cell cycle phasing in Figures 1 and 2. Could you elaborate on how cells were designated by cell cycle phase, and the process by which G2-prophase cells were included or excluded in the quantitative analyses? I believe R1's second comment about the functionality of the tagged CENP-C construct can probably be addressed in the text without additional experimentation (unless you already have in hand experimental data for HA-CENP-C rescue in a CENP-C mutant).

If you could address these two points, we should be able to move forward with a decision on the manuscript. In the revision, please outline how (and where in the text or figures) you have addressed R1's two comments.

In addition we ask that you upload a Striking Image with a corresponding caption to accompany your manuscript if one is available (either a new image or an existing one from within your manuscript). If this image is judged to be suitable, it may be featured on our website. Images should ideally be high resolution, eye-catching, single panel square images. For examples, please browse our archive. If your image is from someone other than yourself, please ensure that the artist has read and agreed to the terms and conditions of the Creative Commons Attribution License. Note: we cannot publish copyrighted images.

We hope to receive your revised manuscript within the next 30 days. Hopefully these minor revisions can be made in short order. If you anticipate a delay in resubmission, we would ask you to let us know the expected resubmission date by email to plosgenetics@plos.org.

[LINK]

Best,

Beth

------

Beth A. Sullivan, PhD

Associate Editor

PLOS Genetics

John Greally

Section Editor: Epigenetics

PLOS Genetics

Reviewer's Responses to Questions

**Comments to the Authors:**

Reviewer #1: Therevised manuscript now entitled ’ CENP-C functions in centromere assembly, the maintenance of CENP-A asymmetry and epigenetic age in Drosophila germline stem cells.’ By Carty et al., has improved the manuscript and the authors have addressed most of the concerns I raised in the initial revision.

However, I still have some issues with Figure 1 and 2, and the way the authors define cell cycle stages. The EdU staining is very obvious and clear. In my view, the other cells are everything but S or M phase (M = condensed chromosomes). The one or two other GSCs in the images that aren’t encircled also fall into the non-S-phase and dot-like spectrosome category for the most part but in, for instance images Figure 1C and C’, have virtually undetectable CENP-C levels. I, therefore, do not see how the authors can quantify a subset of these cells in Figure 1H and conclude that the non-EdU cells that they have encircled are G2-prophase cells. Perhaps I am missing something but with the images and explanations provided this is not obvious to me.

There is also one more concern with the HA-CENP-C overexpression. It does not affect CID assembly and does not rescue the germ line tumor phenotype. The results that overexpression shows a rescue of the differentiation defects and GSC loss (Fig 4) suggests that it is functional but it would be nice to see this with a CENP-C mutant. Perhaps this has been done by the source of the flies (McKim lab). There are many examples of tagged proteins that localize properly but are non-functional in rescue experiments. If this isn’t possible the authors should include a statement that they cannot fully rule out that the tagged protein is fully functional.

Reviewer #2: The authors provided new data and new analyses. In particular, the results in anaphase and telophase GSCs shown in Figure 3 are quite informative. Even though some of the experiments cannot be performed due to pandemic, the revision addresses most of the raised questions.

Reviewer #3: The authors have done an excellent job revising the manuscript. For example, they have modified some figures, been more cautious with some conclusions, added some data, and overall have submitted an improved manuscript.

**Have all data underlying the figures and results presented in the manuscript been provided?**

Reviewer #1: Yes

Reviewer #2: Yes

Reviewer #3: Yes

PLOS authors have the option to publish the peer review history of their article (what does this mean?). If published, this will include your full peer review and any attached files.

Reviewer #1: No

Reviewer #2: No

Reviewer #3: No

---

## [Editor Report · Decision Letter 2]

16 Apr 2021

Dear Dr Dunleavy (Hi Elaine),

Thank you for submitting the final revision of your manuscript and for clarifying the two remaining points raised by Reviewer 1. We are pleased to inform you that your manuscript entitled "CENP-C functions in centromere assembly, the maintenance of CENP-A asymmetry and epigenetic age in Drosophila germline stem cells." has been editorially accepted for publication in PLOS Genetics. Congratulations!

Best wishes,

Beth

---

Beth A. Sullivan, PhD

Associate Editor

PLOS Genetics

John Greally

Section Editor: Epigenetics

PLOS Genetics

Comments from the reviewers (if applicable):

**Data Deposition**

http://datadryad.org/submit?journalID=pgenetics&manu=PGENETICS-D-20-01765R2

**Press Queries**

---

## [Editor Report · Acceptance letter]

28 Apr 2021

PGENETICS-D-20-01765R2 

CENP-C functions in centromere assembly, the maintenance of CENP-A asymmetry and epigenetic age in Drosophila germline stem cells. 

Dear Dr Dunleavy, 

We are pleased to inform you that your manuscript entitled "CENP-C functions in centromere assembly, the maintenance of CENP-A asymmetry and epigenetic age in Drosophila germline stem cells." has been formally accepted for publication in PLOS Genetics! Your manuscript is now with our production department and you will be notified of the publication date in due course.

With kind regards,

Andrea Szabo

PLOS Genetics

On behalf of:
